# Skin Cancer Research Goes Digital: Looking for Biomarkers within the Droplets

**DOI:** 10.3390/jpm12071136

**Published:** 2022-07-13

**Authors:** Elena-Georgiana Dobre, Carolina Constantin, Monica Neagu

**Affiliations:** 1Faculty of Biology, University of Bucharest, Splaiul Independentei 91–95, 050095 Bucharest, Romania; neagu.monica@gmail.com; 2Immunology Department, “Victor Babes” National Institute of Pathology, 050096 Bucharest, Romania; caroconstantin@gmail.com; 3Pathology Department, Colentina Clinical Hospital, 020125 Bucharest, Romania

**Keywords:** ddPCR, skin cancer, biomarkers, liquid biopsy, cutaneous melanoma, squamous cell carcinoma, immunotherapy, targeted therapy, personalized medicine

## Abstract

Skin cancer, which includes the most frequent malignant non-melanoma carcinomas (basal cell carcinoma, BCC, and squamous cell carcinoma, SCC), along with the difficult to treat cutaneous melanoma (CM), pose important worldwide issues for the health care system. Despite the improved anti-cancer armamentarium and the latest scientific achievements, many skin cancer patients fail to respond to therapies, due to the remarkable heterogeneity of cutaneous tumors, calling for even more sophisticated biomarker discovery and patient monitoring approaches. Droplet digital polymerase chain reaction (ddPCR), a robust method for detecting and quantifying low-abundance nucleic acids, has recently emerged as a powerful technology for skin cancer analysis in tissue and liquid biopsies (LBs). The ddPCR method, being capable of analyzing various biological samples, has proved to be efficient in studying variations in gene sequences, including copy number variations (CNVs) and point mutations, DNA methylation, circulatory miRNome, and transcriptome dynamics. Moreover, ddPCR can be designed as a dynamic platform for individualized cancer detection and monitoring therapy efficacy. Here, we present the latest scientific studies applying ddPCR in dermato-oncology, highlighting the potential of this technology for skin cancer biomarker discovery and validation in the context of personalized medicine. The benefits and challenges associated with ddPCR implementation in the clinical setting, mainly when analyzing LBs, are also discussed.

## 1. Introduction

Skin cancer is the most common neoplasm worldwide and is still imposing significant challenge for clinicians and researchers [1]. The different types of skin cancer are named after the cells they originate from, with the most common, and well-characterized, being basal cell carcinoma (BCC), squamous cell carcinoma (SCC) (together referred to as non-melanoma skin cancers (NMSCs)), and cutaneous melanoma (CM) [2]. NMSC accounts for approximately 90% of cutaneous tumors [3]. NMSCs are generally curable and rarely result in death or metastatic disease but can be locally destructive when treatment is inadequate or delayed. In contrast, CM, which encompasses only 1% of skin cancers, is the most life-threatening skin tumor, accounting for about 90% of skin-cancer-associated deaths [4]. According to the most recent GLOBOCAN estimates, there were more than 320,000 new cases of CM worldwide in 2020, which resulted in 57,000 deaths and about 1.2 million new cases of NMSC [5]. However, the latter might be a gross underestimation of the real number, due to challenges related to NMSC diagnosis and reporting [6]. The increased rate of fatality in skin cancer is mainly attributed to late diagnosis, due to non-specific symptoms [7,8], the absence of effective screening methods [9], the lack of sensitive and specific biomarkers for early diagnosis, prognosis, and treatment follow-up [10], as well as to a limited understanding of drug resistance mechanisms in these tumors [11]. Hence, in the last two years, the COVID-19 pandemic, which has become the epicenter of daily clinical practice, restricted access to healthcare facilities and delayed the diagnosis of patients with CM and other skin cancers, resulting in increased rates of morbidity, mortality, and, consequently, a greater financial burden on the health system [12]. Given the poor prognosis of advanced-stage skin cancers, there is an urge to find more reliable biomarkers for early diagnosis, prognosis, and treatment response in these patients.

Recent advances in genetics and bioinformatics technologies revealed dysregulated signaling pathways specific to different cutaneous tumors. While BCC development is linked to deregulation of the Hedgehog (Hh) pathway, SCC and CM are associated with higher tumor mutation burdens and elevated neoantigen load [13]. Although early screening of skin cancers uses improved technologies for dermatologists [14,15,16], complex biomarkers evaluation is needed to prognosticate and individualize the therapy. As a result of an improved understanding of this disease’s biology, the oncological treatment of skin tumors has radically changed over the past decade, switching from a “one-fits-all” chemotherapeutic approach to a more tailored perspective, where therapies are only given when particular patient- and tumor-specific features are present [17]. For instance, the latest progress in understanding the Hedgehog (Hh) signaling pathway and its role in BCC pathogenesis, led to the development of pathway inhibitors vismodegib and sonidegib, which have considerably transformed the clinical management of metastatic BCC [18]. Additionally, genomic profiling of CMs revealed several actionable mutations, such as BRAF V600E/K, that can be triggered with specific BRAF inhibitors (BRAFi: vemurafenib and dabrafenib) or MEK inhibitors (MEKi: trametinib, binimetinib, and cobimetinib), resulting in improved overall response (OR) and overall survival (OS) rates in CM metastatic patients [19]. Furthermore, the discovery that the CM and SCC genomes are highly mutagenized, with a high load of neoantigens, highlighted the immunogenic nature of these entities and enabled the development of immune-checkpoint inhibitors (ICIs), such as Pembrolizumab (anti-PD1), Nivolumab (anti-PD-1), Cemiplimab (anti-PD-1) and Ipilimumab (anti-CTLA4) [13,20]. Although immunotherapies and targeted therapies have greatly improved the pathological complete response (pCR) and OS in skin cancer patients, their efficiency is often limited by the increased genomic and immune heterogeneity of tumors, calling for even more refined approaches for skin cancer treatment and monitoring [21,22].

Personalized medicine, the future proposed for cancer care, is based on a tailored approach that selects the most appropriate therapy for each cancer patient, considering its unique molecular features [23]. The personalized approach involves a complete biochemical characterization of the tumor using multi-dimensional analyses for a range of biological endpoints, standing for molecule-level cancer re-classification to evaluate the metastatic potential and deliver the most promising treatment [24]. Among the omics technologies that gained momentum in the last years, droplet digital polymerase chain reaction (ddPCR) offered the possibility of accurately detecting and quantifying low-abundance nucleic acids in various biological samples, having important applications in cancer subtyping [25], prognosis [26], and minimal residual disease monitoring [27]. Suitable for both archived and liquid biopsy (LB) samples, ddPCR can be used for numerous omics measurements, including absolute allele quantification [28], CNVs analysis [29], rare mutations [30], DNA methylation detection [31], transcriptomic evaluations (mRNA, miRNA) [32] and genomic rearrangements [33]. Therefore, ddPCR forms a suitable platform to be used in personalized medicine in oncology. Although it is a robust omics technology, ddPCR is unsuitable for genome-wide, exploratory measurements [34]. Yet, it is more appropriate for analyzing a small number of known markers and mutations, which is why it is almost always coupled with whole-genome profiling technologies, such as Next Generation Sequencing (NGS). Being more practical and affordable, ddPCR may be employed to turn the putative biomarkers discovered via NGS into valuable indicators of tumor progression and therapy effectiveness in cancers [35]. Considered the third generation of PCR, ddPCR divides the nucleic acid sample into thousands, or even millions, of droplets that serve as separate reaction chambers [36]. This partitioning process brings several improvements over traditional PCR techniques, that consist in absolute quantification of the target sequence, without the need for calibration and internal controls used in Real-Time Quantitative (q-)PCR, greater tolerance to inhibitors, and increased accuracy when working with low target concentrations or degraded samples [37].

Due to its versatility and ability to operate with small amounts of biological material, ddPCR is an ideal methodology for analyzing LBs in cancers [30]. LB, based on the analysis of cancer-derived components, such as circulating tumor DNA (ctDNA), RNA, extracellular vesicles (EVs), circulating tumor cells (CTCs), and tumor-educated platelets (TEPs) in the biofluids of patients, has gained considerable attention, due to its potential to provide relevant information about tumor evolution and therapeutic responses in real-time [38]. Therefore, LB emerged as a complementary non-invasive method to surgical biopsy, overcoming the recurrent limitations associated with the clinical assessment of inaccessible tumors and their clonal heterogeneity [39]. Several blood-based biomarkers interrogated by ddPCR have already found diagnostic [40], predictive [41], and monitoring purposes [42] in certain tumor types. Furthermore, ongoing ddPCR approaches are oriented towards harnessing other body fluids, such as cerebrospinal fluid or urine, to find reliable biomarkers for cancer patients [43,44]. Recently, in CM, ddPCR proved to be a reliable tool to quantify low-abundance point mutations in plasma ctDNA to reflect intra-tumoral heterogeneity and to track the dynamic changes in tumor burden after treatment exposure during follow-up [27].

Here, we review the latest scientific results obtained from research applying ddPCR in the field of dermato-oncology. We highlight how ddPCR, a relatively young omics technology, can help identify potential candidate biomarkers for diagnosis, prognosis, and screening of skin cancers, as well as putative therapeutic targets, forming a platform for personalized medicine in oncology.

## 2. The ddPCR Method: A Reliable Omics Technology in Oncology

The ddPCR method is a significant development of PCR technology that has considerably changed life science research and molecular diagnostics. Early attempts to set up the procedure and optimize it were described in the 1990s when various research groups applied limiting dilution conditions to obtain single PCR molecules [36]. A considerable advantage of limiting dilution PCR is that each DNA molecule may be amplified separately to reduce template competition during PCR and the background noise in complex samples [36]. In 1992, Sykes et al. were the first who combined the principles of limiting dilution, PCR, and Poisson statistics to quantitate the total number of rare leukemic cells in an excess background of normal leukemic cells [45]. At that time, other research groups employed versions of limiting dilution PCR to examine variations among HIV proviruses [46] and human genomic haplotypes [47], as well as to quantify the fraction of leukemic cells after chemotherapy [48].

Yet, the term “digital PCR” was first introduced in 1999, when Vogelstein and Kinzler described a new method to quantify disease-associated mutations in biological samples from colorectal cancer (CRC) patients [49]. Their methodology consisted in applying the dilution PCR strategy (into a 384-well plate) to enable the separate amplification of the individual template molecules so that the resultant PCR products could either be completely mutant or completely wild-type (WT). Finally, PCR partitions were read and counted as negative or positive by fluorescence analysis, enabling the quantification of target molecules under the assumption of Poisson distribution [49]. Considering the classification of the reactions as ‘‘negatives/zeros’’ or ‘‘positives /ones’’, Vogelstein and Kinzler termed their method “digital PCR” by analogy to the binary code used in computer science [50]. Interestingly, their study documented a variety of possible applications of ddPCR in oncology, including detecting SNVs, changes in gene expression, alternatively spliced products, chromosomal translocations, and allelic discrimination in tumors [49]. Furthermore, they emphasized that, due to its ability to accurately detect cancer-causing somatic mutations in an excess of WT DNA, ddPCR could help diagnose primary tumors in asymptomatic patients when the disease is still curable, providing an advantage for early diagnosis and preventive medicine in cancer [49].

In the early 2000s, advances in microfluidics and informatics, coupled with novel water-in-oil emulsion systems for sample partitioning, have allowed the development of more sophisticated equipment capable of subdividing PCR reactions into smaller reaction volumes [51]. Since then, different digital PCR platforms have been released, and are nowadays available, including the microfluidic chamber-based Biomark^®^ system from Fluidigm [52], micro-well chip-based Quantstudio 12k/3D^®^ from Thermo Fisher Scientific [53], droplet-based QX100 and QX200 Droplet Digital PCR^®^ from Bio-Rad [30], the RainDrop dPCR^®^ from RainDance technologies [54], the Crystal digital PCR^®^ from Stilla Technologies [55], the BEAMing^®^ technology from Sysmex Inostics [56], the Lab On An Array (LOAA) Digital PCR^®^ system from Optolane [57] and the QIAcuity Digital PCR^®^ system from Qiagen [58]. These systems have proven suitable for cancer research, showing similar results in terms of nucleic acid quantification, specificity and sensitivity. Among them, the Bio-Rad ddPCR platform is the most popular and extensively used, due to its remarkable accessibility and adaptability, being less time-consuming and laborious compared to classical methods [59,60]. In our review, we focus on ddPCR as a reliable tool for analyzing and monitoring skin cancers.

The ddPCR method involves a massive sub-partitioning of the nucleic acid sample into 20,000 nanoliter-sized droplets that serve as separate tubes or reaction chambers [61]. Droplets are generated in a water-in-oil emulsion and stabilized using proprietary PCR-compatible surfactants [59]. Technically, the nucleic acid sample (with the ddPCR Master Mix, primers, and probes in a final volume of 20 μL) is dispersed to 1 nL-droplets in an oil phase when passing through the microfluidic channels of a droplet generator cartridge [59] (Figure 1). Subsequently, the droplets are transferred from the 8-sample cartridge into a 96-well plate for PCR amplification [62]. Each droplet contains one or no copies of the target sequence. Following PCR amplification of the nucleic acid target in the droplets, the samples are analyzed by a Droplet Reader (Figure 1) [62]. Here, each droplet is examined individually for amplified DNA using a two-color fluorescence detection system (set to detect FAM and either HEX/VIC), and the number of positive and negative droplets are used to calculate the concentration of the target sequence, applying an analysis method based on Poisson distribution [63]. The partitioning process brings several improvements over the traditional PCR techniques that consist in absolute quantification of the target sequence without the need for calibration and internal controls used in qPCR. The improvements include greater tolerance to inhibitors, increased accuracy when working with low target concentrations or degraded samples, as well as remarkable sensitivity and repeatability of the experiments [37].

The qPCR has to date represented the method of choice for quantitative gene expression measurements in biological samples. Still, the resulting data can be highly heterogeneous, artifactual, and non-reproducible, requiring an accurate verification and validation of both samples and primers. As a method, qPCR relies on real-time monitoring of the fluorescence increase per cycle during the amplification of DNA [37]. Gene expression data generated using qRT-PCR can be analyzed by two approaches, absolute or relative quantification. Absolute quantification links the PCR signal to the input copy number using a calibration curve, whereas relative quantification measures the change in mRNA expression levels by employing an internal calibrator, a constitutively expressed transcript commonly referred to as a “housekeeping” gene [64]. The major limitation of the absolute method is its inability to account for any procedure that may introduce inter- or intra-sample variability [65]. The quantification cycle, or Cq value, of an amplification reaction is defined as the fractional number of cycles needed for the fluorescence to reach a quantification threshold. Optimization is critical in qPCR for each primer set, so that reaction efficiency is consistent between all samples, with sample contaminants appropriately diluted to ensure that all reactions and associated Cq values are within the validated analytical range of the respective standard curves [37]. However, the ddPCR concept has brought many benefits over real-time PCR. Although both techniques employ Taq polymerase to amplify target DNA sequences from complex samples, the partitioning step in ddPCR offers the advantage of direct and independent quantification of DNA without standard curves, generating more accurate and reproducible data compared to qPCR assays, especially in the presence of sample contaminants that can partially inhibit Taq polymerase and primer annealing [37]. QPCR can distinguish between CNVs or differences in gene expression that are two-fold or greater [37]. Nonetheless, ddPCR enables accurate quantification of expression differences that are two-fold or lower, identifies alleles that are less frequent than 0.1%, and distinguishes between copy number variations that are different by only one copy [59]. In samples with low concentrations of nucleic acids and variable amounts of inhibitors, ddPCR technology was shown to convert uninterpretable results generated from qPCR to highly quantitative and reproducible data [37]. Hence, in ddPCR, the analysis results are directly expressed as number of copies of target per microliter of reaction (with confidence intervals), significantly reducing the bias the operator may introduce during the data processing step [66]. These findings demonstrate that ddPCR offers improved analytical sensitivity and specificity for CNVs and gene expression measurements, being suitable for various molecular applications in oncology.

To develop a fully robust ddPCR assay, several analytical parameters, such as Limit of Blank (LoB), Limit of Detection (LoD), and Limit of Quantitation (LoQ), should be determined. These parameters define the quality of a ddPCR assay [36]. LoB is the highest apparent target concentration expected to be found when replicates of a blank sample containing no target sequences are tested. In contrast, LoQ represents the lowest concentration at which the analyte can be quantified [67]. LoD is the lowest target concentration likely to be reliably distinguished from the LoB (95% detection limit or a type II error of 5%) and at which detection is feasible [67]. Notably, the LoD of ddPCR is about 0.005%, below that of RT-PCR (1%), pyrosequencing (5%), melting curve analysis (10%), and Sanger sequencing (20%) [68]. Furthermore, ddPCR presents an increased sensitivity, ranging from 0.001% to 0.1%, suitable for minimal residual disease (MRD) monitoring approaches [35]. However, the maturation of ddPCR methods into reliable analytical methods ideal for diagnostics and other clinical purposes requires these methods to be validated for their intended use. The aforementioned analytical parameters remain one of the most critical performance characteristics to be assessed during method validation, according to the international standards ISO/IEC 17025 and ISO 15189 [69]. Detailed requirements for method validation can also be found in guidelines on Minimum Information for the publication of Quantitative dPCR Experiments (dMIQE) [70,71]. Among the enclosed recommendations for the standardization of experimental protocols, the need for appropriate quality controls is particularly emphasized. For instance, negative controls may help detect the false-positive reactions that may occur as a consequence of cross-contamination between samples, as well as from non-specific binding of probes and primer dimer formation [71]. Furthermore, ddPCR requires a threshold to distinguish positive from negative partitions and determine the false-positive and false-negative rates, which impacts the validity and accuracy of the ddPCR assay. Moreover, the assays aiming to detect rare variants should also include a WT control containing the WT sequence. Hence, when designing ddPCR assays for rare event detection, controls with different known proportions of WT and mutant sequences are strongly recommended [70].

Nowadays, minimally invasive technologies, such as LBs capturing tumor markers in body fluids, hold great promise for personalized cancer treatment, due to their ability to provide multiple non-invasive global snapshots of primary and metastatic tumors. Currently available technologies for ctDNA analysis are based on PCR and NGS. However, when we talk about personalized cancer care, the need arises to use the most reliable technology, which should be accurate and advantageous in terms of economic resources and time, as well as ease of use [35]. While ddPCR remains a tumor agnostic, cheap, and with a quick turnaround, there are several limitations really important to consider in terms of wider usage. In the first instance, ddPCR requires a priori knowledge of specific genetic alterations, being unsuitable for genome-wide exploratory studies [35]. Several sets of primers and probes targeting specific genomic regions can be mixed in one PCR reaction to generate multiplex PCR. This multiplexing, however, comes with many challenges, such as varying efficiency of individual assays, different primer annealing temperatures, possible oligonucleotide cross-dimerization, and inaccurate separation of fluorescent signals from a specific panel of reporter dyes with overlapping emission spectra. Therefore, analyzing broader genomic regions using ddPCR may not be possible [72]. Furthermore, ddPCR comes with the limitation of not ensuring a consistent and accurate detection of low-frequency variants in LB samples. It has been mainly noted in studies on pancreatic ductal adenocarcinoma (PDAC), where a significant percentage of patients display KRAS mutations at a low allele frequency. Despite the considerably high sensitivity of digital PCR, the detection of KRAS mutations in the plasma of PDAC patients using this method has failed expectations, as the ctDNA detection rate was reported to be as low as 50% [73]. Moreover, when operating with ctDNA, PCR assays face challenges, due to a lack of specificity, and can result in high false-positive rates. For a given assay, the relative fluorescence signal that discerns a true positive droplet from a negative droplet can vary greatly dependent on multiple factors, including template sequence, quality of the amplicon, polymerase-induced errors, cycling conditions employed, concentrations of key reagents and instrument artifacts [72]. Input material quantity and quality are also critical, as amplification steps cannot replace the low input of cfDNA; therefore, the polymerase will introduce errors, increasing the risk of having false-positive variants [74]. Hence, for assays intended to test for multiple mutations at once in a sample, these challenges are multiplied, and the performance of each assay will be different between single-plex and multi-plex, even if operating with the same reagents at the same concentrations and under the same reaction conditions [72].

NGS technologies are addressing these shortcomings and may fulfill the promise of personalized medicine as researchers obtain valuable multi-omics data on tumor material. Advances in NGS are leading to marked improvements in the accuracy and detection limit of LBs and the potential number of measurable biomarkers per assay. Due to its high-throughput and massive parallel sequencing capabilities, NGS can screen for various omic alterations (e.g., rare mutations, mRNA expression, DNA methylation, gene amplification, and gene fusions) with either prognostic or therapeutic potential in multiple samples simultaneously [75]. Several NGS methods have been developed for detecting ctDNA and these are subdivided into two groups, namely, targeted and untargeted strategies. Targeted approaches focus on detecting specific alterations in a batch of predefined genes. Typical examples of targeted NGS methods for quantifying genetic mutations for different cancers include tagged-amplicon deep sequencing (TAm-seq), safe-sequencing system (Safe-SeqS), and cancer personalized profiling by deep sequencing (CAPP-Seq). Targeted NGS approaches proved extremely sensitive, as mutations can be detected at an allele frequency of down to 0.01% with high specificity and sensitivity [76]. In contrast, untargeted NGS approaches aim at a genome-wide analysis for CNVs or point mutations by whole-genome sequencing (WGS) or whole-exome sequencing (WES). Although untargeted approaches can identify novel changes occurring during tumor treatment without requiring prior information about the primary tumor mutational landscape, they are less sensitive than targeted strategies [76]. Discriminating between true mutations and false-positive variants remains a major challenge in NGS. Still, within the last years, library preparation protocols have been upgraded to improve the detection of rare variants [74]. Nonetheless, NGS technology is not suitable for patient longitudinal monitoring, as it is expensive, meticulous, and requires powerful bioinformatics support. Yet, coupling NGS with ultrasensitive ddPCR may help overcome the limitations and increase the benefits of both techniques. Therefore, once a specific panel of genomic alterations has been identified via NGS, researchers can use ddPCR, which is less costly and laborious, to assess that set of biomarkers and get important information on the course of the disease and therapeutic responses [77]. The strengths and limitations of NGS, ddPCR and other conventional technologies currently employed for routine molecular genetic tumor testing are summarized in Table 1.

## 3. The ddPCR Method for Primary Prevention Strategies and Personalized Skin Cancer Screening

One of the prospective applications of ddPCR in skin cancer research may be in the field of preventive medicine. Established risk factors for skin cancer include environmental carcinogens, such as ultraviolet (UV) light exposure, immunosuppression, chronic inflammation, genetic background, and infection with certain human papillomavirus (HPV) genotypes [79]. A bourgeoning body of research highlights an etiologic relationship between HPV infection and skin cancer, particularly between types in genus β and SCC [80,81]. By efficiently deregulating the p53 and pRB tumor-suppressor pathways, oncoproteins E6 and E7 from β-HPV may cause interferences in the cell cycle, leading to the immortalization of the HPV-infected cells [79]. A recent study with 1008 participants conducted by Rollins et al. has shown that the presence of β-HPV at the baseline, particularly in the skin swabs, significantly predicted the development of SCC (HR = 4.32; 95% CI, 1.00–18.66), whereas serologic evidence of past β-HPV infection was not linked with the SCC risk [82]. Therefore, type-specific HPV-DNA detection by the ultrasensitive methodology of ddPCR may be a valuable strategy for identifying individuals at higher risk for SCC, holding promise for improved keratinocyte cancer prevention and screening initiatives aimed at minimizing the incidence, morbidity, and economic burden associated with such a diagnosis. ddPCR has recently proved its feasibility and accuracy in detecting HPV-16/HPV-18-DNA in formalin-fixed paraffin-embedded (FFPE) tissues from patients with oropharyngeal squamous cell carcinomas (OPSCCs), strengthening the clinical relevance of p16-immunohistochemistry (p16-IHC) status in this disease [83].

Furthermore, ddPCR may be a valuable tool for assessing heritable mutations that may increase the risk of NMSC and CM. Interestingly, patients with xeroderma pigmentosum harboring mutations in nucleotide excision repair (NER) genes have a 1000-fold higher risk for cutaneous malignancies than the general population [84]. Moreover, patients with basal cell naevus syndrome (BCNS)/Gorlin-Goltz syndrome displaying germline mutations in the human homolog of the Drosophila patched-1 gene (*PTCH1*) are prone to develop multiple BCCs during their lifetime [85]. The incidence of BCNS is estimated at 1 in 56,000–256,000 individuals [86]. BCNS diagnosis is usually based on clinical criteria, considering the presence of multiple odontogenic cysts and skeletal abnormalities, a calcified falx cerebri, and an increased number of cutaneous nevi; however, it may also require genetic confirmation in some cases, particularly in children or in patients with postzygotic mosaicism (in *PTCH1* or *SMO*) [85]. In post-zygotic mosaicism, a mutation usually occurs early in embryogenesis, affecting only cells of a mutant progenitor, leading to a mixture of healthy and affected cell populations. Depending on the tissues involved in mosaicism and the mutational load, the clinical manifestations may be more or less visible in individuals with post-zygotic mosaicism [87]. Interestingly, Reinders et al. have recently employed ddPCR to confirm that low-grade postzygotic mosaicism of *PTCH1* gene mutation may induce clinical manifestations similar to those caused by a germline mutation in a BCNS-suspected patient [87]. The patient Reinders and his team analyzed also developed a BCC on the left cheek during the dermatological follow-up. According to the percentages found with ddPCR in different tissues, the degree of gonadal mosaicism for the analyzed patient was somewhere between 13% and 17% [87]. If the degree of gonadal cells with the *PTCH1* gene mutation is below 50%, transmission to the offspring would probably be lower than 50%. Still, it should be considered when asking for informed decisions about prenatal or preimplantation genetic diagnosis [87]. Taken together, all this information suggests that genetic testing, assisted by ddPCR, should be performed for all the patients suspected of BCNS, even if they do not meet the clinical criteria, as it may have considerable implications for skin cancer prevention and genetic counseling in affected patients.

Comparative genome-wide studies revealed significant differences in mutational profiles between hereditary and sporadic melanomas. *CDKN2A* (encoding p16INK4a and p14ARF) and *CDK4* are the major high-penetrance susceptibility genes, with germline mutations identified in 20–40% of high-risk families [10,88]. In contrast, sporadic CMs carry mutations in the *BRAF*, *NRAS*, and *NF1* genes [89]. These alterations occur early in carcinogenesis, may coexist, and result in the constitutive activation of the oncogenic mitogen-activated protein kinase (MAPK) pathway [90]. Besides their roles in the early diagnosis of CM, hotspot mutations such as *KRAS* and *BRAF* are also regarded as potential therapeutic targets for pharmacological interventions in clinical management [19]. Although emerging targeted therapies, such as the *BRAF* inhibitors vemurafenib and dabrafenib, improve prognosis, they require an accurate and sensitive detection of the hotspot oncogenic mutations [91]. Notably, ddPCR showed the highest sensitivity in detecting the *BRAF* V600E mutations in FFPE tissues harvested from 87 CM patients with Breslow stage I-V disease, among four technologies actively employed in the clinical setting to assess it: the Cobas^®^ 4800 system, based on real-time PCR amplification, Sanger sequencing, and the allele-specific PCR (AS-PCR) (35.6% vs. 9.2%, 26.4%, and 26.4%) [91]. For eight patients in the clinical cohort, the *BRAF* V600E mutation was only detectable by ddPCR; therefore, all these patients would have been eligible for vemurafenib therapy. Hence, five out of these eight patients who tested *BRAF* V600E positive only through ddPCR presented later with sentinel lymph node metastases, suggesting that ddPCR should be the primary approach for detecting and monitoring *BRAF* V600E-mutant melanomas [91]. In line with these observations, McEvoy et al. demonstrated that ddPCR is more accurate than pyrosequencing and Sanger sequencing in detecting common *BRAF*, *NRAS*, and *TERT* promoter mutations in 40 FFPE melanoma tissues [68]. DdPCR identified hotspot mutations in 12.5% and 23% of tumors deemed as WT by pyrosequencing and Sanger sequencing and hence showed an excellent sensitivity when analyzing tumors with <50% tumor cellularity [68]. Therefore, the implementation of ddPCR-based assays in CM can revolutionize the clinical management of this disease, as it may facilitate the analysis of early-stage tumors and support research into improving outcomes in melanoma patients. 

Nonetheless, regarding the use of *BRAF* V600E mutation as a biomarker for CM early detection, several precautions should be considered before its implementation in clinical practice. According to the literature, the *BRAF* V600E mutation is a common event not only in acquired benign and dysplastic nevi, but also in congenital nevi. However, the majority of nevi do not progress to melanoma [92]. *BRAF* mutations are frequent in melanocytic nevi and vertical growth phase melanomas but infrequent in the radial growth phase and in situ melanomas. Thus, while *BRAF* mutations undoubtedly drive melanoma growth and progression, they are insufficient by themselves to induce melanomas [19]. These findings indicate the controversy of the theory that *BRAF* oncogene activation is a crucial early event in melanoma progression. Additionally, they highlight some of the complexities underlying melanomagenesis and the need for further understanding the relationship between *BRAF* and other mutations before validating *BRAF* V600E mutation as a biomarker for early CM detection.

## 4. The ddPCR Nethod Assisting the Prognosis of Skin Cancer

At the moment, the skin cancer staging system relies on the assessment of clinicopathological variables, such as the size of the primary tumor (T), dissemination in the lymph nodes (N), and distant metastasis (M) [93]. Besides their role in tumor staging, clinicopathological variables (e.g., Breslow thickness, ulceration, Clark level of invasion, mitotic rate, and regional lymph node status) are also valuable indicators of skin cancer prognosis. Histopathologically, Breslow’s thickness, which represents the distance between the cutaneous surface (granulous layer) and the deepest point of tumor penetration, remains one of the most important prognostic factors for metastases in CM. According to the Breslow index, CMs may be stratified into thin lesions (less than 1 mm thick), intermediate lesions (1–4 mm thick), and thick lesions (more than 4 mm thick) [94]. Thin lesions are treated through surgery and have an almost 100% (depending on the presence of ulceration) 5-year survival rate in Australia. Hence, survival decreases considerably with every millimeter increase in thickness and drops to 54% for CM tumors greater than 4 mm [95]. Regional lymph node status remains another important prognostic indicator in early-stage melanoma. Timely evaluation of the regional lymph node basin with sentinel lymph node (SLN) biopsy can identify clinically occult lymph node metastases, allowing early therapeutic lymph node dissection in order to prevent cancer’s spread [96]. According to the latest AJCC edition, SNB should be considered for patients with T1b melanomas of thickness 0.8 to 1.0 mm or less than 0.8 mm Breslow thickness with ulceration, classified as T1b lesion [97]. A positive SN has been reported in approximately 5.2% of thin melanomas and 8% of CMs thicker than 0.8 mm. Moreover, ulceration is associated with increased risk for SN positivity, while there is little supporting evidence that mitoses in thin melanomas are associated with SN positivity [97]. However, clinicopathological variables are not relevant to the increased genetic and immune heterogeneity of skin cancers and, therefore, may not be informative about the prognosis and clinical outcome of the disease in certain cases [21]. Thus, it is crucial to construct a prognosis gene signature more reliable than the traditional TNM staging system to identify high-risk skin cancer patients. 

Recently, the mutational status of the human telomerase reverse transcriptase (hTERT) gene emerged as an important indicator of diagnosis and prognosis in CM [98]. The reactivation of telomerase, which ensures the replicative immortality of human cancers, is thought to be driven by transcriptional upregulation of the hTERT gene [99]. However, although hTERT promoter (hTERTp) mutations are the primary cancer-associated genetic mechanism of *TERT* upregulation, additional genetic and epigenetic events may also contribute to hTERT upregulation in cancer cells [100]. Recently, Salgado et al. employed ddPCR technology to investigate the molecular mechanisms responsible for hTERT reactivation in CM [101]. Besides two hotspot mutations in the TERTp, dubbed C228T and C250T, which are notorious for their involvement in *TERT* mRNA upregulation, the authors also reported hTERTp hypermethylation in the analyzed samples [101]. Subsequently, they developed a ddPCR protocol to assess TERTp methylation fraction (MF) alongside C228T and C250T TERTp mutations in 44 healthy, benign and malignant tumor samples. They noticed that hTERT expression depends on TERTp methylation and chromatin accessibility in the human melanoma cell lines they analyzed; therefore, in the case of TERTp-wild type samples, *TERT* expression required an open chromatin state due to increased TERTp methylation; hence, in the case of C228T/C250T-positive samples, hTERT expression involved a combination of moderate MF and chromatin accessibility [101]. Given that TERTp mutations and hypermethylation correlate with a poor prognosis and lower overall survival in CM [102], TERTp assessment by ddPCR may guide the development of improved prognostic assays to stratify CM patients according to clinical risk.

In addition to their roles in CM early diagnosis, hotspot mutations, such as *NRAS* and *BRAF*, are currently investigated as putative prognostic biomarkers for this unpredictable disease [103]. About 40–60% of melanocytic tumors harbor activating *BRAF* V600 mutations (over 90% V600E), which are the most commonly found mutations in CM [104]. The second most common genetic aberration in CM is mutated NRAS, occurring in ~20% of cases [105]. Several research groups noted that *BRAF*-mutated melanomas present an increased propensity to metastasize to distant sites, being much more invasive than WT melanomas [103]. *BRAF* V600K variants were reported to be more aggressive than *BRAF* V600E ones since they have a shorter disease-free interval from diagnosis of primary melanoma to the occurrence of first distant metastasis, as well as inferior tumor regression and shorter PFS when treated with *BRAF* and MEK inhibitors [106]. Nonetheless, the data on *BRAF* variant outcomes are based on small numbers in sub-selected populations, hence not widely validated to the extent of making long-term prognostic claims. In addition, treatment outcome data also relies on a post hoc analysis, which may introduce critical selection bias, suggesting that the predictive value of *BRAF* variants remains just speculative until their validation in larger prospectively curated cohorts. *NRAS*-mutant melanomas are much more unpredictable and bear a worse prognosis when compared to melanomas driven by other RAS isoforms or *BRAF*-mutant melanomas [105]; however, a recent study potentiated that brain metastases emanating from thin and un-ulcerated tumors are enriched in *KRAS* mutations, occurring mainly in codons 12, 13 and 61 [107]. Considering the retrospective nature of this analysis, the prognostic value of *KRAS* mutations in melanoma remains suppositional. The identification of *KRAS* mutations as a predictive biomarker for the development of early brain metastases requires prospective validation in larger cohorts employing multivariate models, particularly assessing the predictive value of these mutations in relation to other clinicopathological variables.

Nonetheless, discordant mutational profiles have been reported between different sites of a primary tumor (intra-tumor heterogeneity), between a primary tumor and metastases, and between different metastases of the same patient (inter-tumor heterogeneity) [108]. This is mainly because primary cutaneous tumors are composed of multiple genotypically and phenotypically distinct cell populations [109]. Due to its increased sensitivity and accuracy when operating with low amounts of biological material, ddPCR seems to be the ideal tool for assessing tumor heterogeneity, which may be relevant for investigating therapeutic responses and survival outcomes in skin cancer patients. Interestingly, by using a combination of SNaPshot assays, Sanger sequencing, and ddPCR, Chang et al. evaluated the presence of *TERT* , *BRAF*, and *NRAS* mts in paired primary and metastatic tumors from 60 patients and in multiple metastatic tumors from 39 patients whose primary tumors were unavailable [110]. Overall, they identified mutational heterogeneity in 18 of 99 patients (18%). Among patients with available primary tumors, 12 of 60 displayed mutational heterogeneity between their primary and metastatic tumors. This included some cases in which a new mutation was discovered in one or more metastatic lesions, consistent with disease progression and the emergence of highly mutated tumor genotypes over time [110]. They also reported cases in which mutations identified in primary tumors were undetectable in one or more metastatic tumors, suggestive of poly-clonality in the primary tumor. To address the concern regarding undiagnosed secondary melanomas, they also analyzed the inter-tumor heterogeneity between metastatic tumors from individual patients [110]. They found nine patients with different *BRAF*, *NRAS*, or *TERT* genotypic profiles between their metastatic tumors. Of these nine patients, five displayed additional mutations in metastases that developed at later time points, suggesting that the development of subclones may be inherent in CM even in the absence of the therapeutic pressure [110]. Therefore, ddPCR may be a promising approach for investigating the clonal heterogeneity in melanoma, which contributes to therapeutic resistance, impacts patients’ prognosis, and may be highly relevant for treatment design in this hard-to-treat disease.

Although the identification of robust prognostic biomarkers is a constant concern for researchers and clinicians in dermato-oncology, progress registered in this regard for NMSC is relatively rare [10,111]. Recently, many studies have indicated the promising potential of miRNAs as biomarkers with diagnostic and prognostic applications in BCC and SCC [112,113]. MiRNAs are single-stranded RNA molecules (~22 nucleotides in length) that can fine-tune gene expression at the post-transcriptional level through an interaction with the 3′-untranslated region (3′ UTR) of the target mRNA or through binding with other regions, such as the 5′ UTR, the coding sequence and gene promoters [114]. MiRNAs are critical regulators of various physiological processes; however, miRNAs’ aberrant expression is frequently reported in cancers, where they regulate different tumor biological properties, including invasiveness [115]. Recently, miR-34a has emerged as an important diagnosis and prognosis indicator in BCC [112]. MiR-34a expression is lower in patients having BCC than in healthy controls and correlates with tumor cell diameter, lymph node metastasis, and BCC histological types. Hence, BCC patients harboring low levels of miR-34a carry a poor prognosis [112]. In parallel, Canueto et al. reported that miR-203 and miR-205 expression patterns might be used as determinants of prognosis in SCC patients [113]. While miR-205 has been associated with pathological features of poor prognosis, including desmoplasia, perineural invasion, and infiltrative patterns, miRNA-203 expression was linked to a favorable prognosis, due to its identification, mainly in well-differentiated areas and rarely in the invasion site [113]. As ddPCR is becoming extremely popular for assessing miRNA profiles in various cancer types [116], the ddPCR-evaluation of skin cancer progression-associated miRNAs is soon expected to find its place in clinical practice, as a promising strategy to improve risk stratification and the clinical management of NMSC patients.

## 5. DdPCR-Based Liquid Biopsies for Skin Cancer Monitoring and Post-Treatment Follow-Up

There are many treatment options for skin cancers. Surgical excision is the mainstay of local treatment, whereas radiotherapy (RT) is recommended for patients with medically inoperable, surgically unresectable disease or with high predisposition to metastasis [117,118]. Outcomes are generally inferior with RT without surgical resection and tumors may recur quickly after the treatment [119,120]. Novel systemic approaches, such as targeted therapies and immunotherapies, have revolutionized skin cancer therapy and improved clinical care, especially in the metastatic setting [118,121]. Hedgehog pathway inhibitors (HHi), vismodegib and sonidegib, are relevant examples in this context. These compounds gained US Food and Drug Administration (FDA) approval in 2012 and in 2015 showed increasing efficiency in metastatic BCC patients when tested in phase I/II clinical trials [122,123] and were entered into patients’ clinical management [124]. Other examples are the MAPK pathway-targeting drugs that the FDA has approved for the treatment of nonresectable or metastatic CMs with *BRAF* mutations. There are at least five targeted anti-cancer agents that have gained FDA approval since 2011 until now: the *BRAF* inhibitors (BRAFi: vemurafenib- 2011 and dabrafenib- 2013) and the mitogen-activated protein kinase kinase inhibitors (MEKi: trametinib- 2013, cobimetinib- 2015, and binimetinib- 2018) [125]. Nonetheless, BRAFi and MEKi, either alone or in combination, provide rapid disease control with high response rates in patients with BRAF-mutant metastatic melanoma. However, therapeutic responses achieved with targeted therapies are heterogeneous and not always durable [126]. The discovery that CM and NMSC genomes are highly mutagenized, with a high load of neoantigens, highlighted the immunogenic nature of these entities and their ability to respond to immunotherapies [13,127]. The anti-cytotoxic T lymphocyte antigen (CTLA-4) antibody Ipilimumab (Yervoy™) was the first immune-checkpoint inhibitor receiving FDA approval in 2011 for the treatment of metastatic CM, closely followed by the anti-programmed death (PD-1) antibodies Nivolumab (Opdivo™) and Pembrolizumab (Keytruda™) in 2014 [126]. Notably, the use of these immunotherapies in the clinical setting has considerably improved the clinical evolution of CM patients, leading to more durable results and even pathological complete response (pCR) in some individuals [128]. A scheme outlining the major gene-related biomarkers in skin cancers and their propensity to develop a targeted therapy is presented in Figure 2. Yet, there is increasing evidence that patients with advanced NMSCs may also benefit from the successes of immunotherapies with anti-CTLA-4 and anti-PD-1 monoclonal antibodies [129,130]. However, although immunotherapies and targeted therapies have greatly improved the clinical outcome and OS in skin cancer patients, their efficiency is often limited by the increased genomic and immune heterogeneity of tumors, calling for even more refined approaches for skin cancer treatment and monitoring [21,22].

LB, which relies on the analysis of circulating components derived from the tumors within body fluids, has recently emerged as a powerful tool for monitoring disease evolution and therapeutic responses in skin cancer patients. In contrast to the highly invasive and spatially limited tissue biopsies, LB allows for repeated sampling and can reflect the molecular heterogeneity of cutaneous tumors more exhaustively, containing analytes derived from different areas of the same tumor and possible metastatic sites [131]. The body fluids currently exploited as LBs for clinical non-invasive evaluations include blood, amniotic fluid, pleural fluid, saliva, ascites, urine, and cerebrospinal fluid (CSF) [132]. Hence, the potential applications of molecular and cellular analytes derived from tumors, such as circulating tumor DNA (ctDNA) and RNA, extracellular vesicles (EVs), circulating tumor cells (CTCs), and tumor-educated platelets (TEP), are outstanding and are related to early diagnosis, prognosis, and monitoring of the disease at the molecular level during treatment [131,132]. Due to its increased sensitivity and sensibility when operating with low-input samples, ddPCR has become one of the most popular tools for examining omics alterations in LBs in human cancers [133]. Here, we describe the ddPCR-based LB applications in longitudinal monitoring of skin tumors under treatment as an important strategy to detect disease recurrence prior to clinical symptoms or imagistic evaluation. 

### 5.1. CtDNA Analysis

One of the most important applications of ddPCR relies on assessing circulating tumor DNA (ctDNA) in skin cancer patient-derived LBs to quantify the dynamic changes in tumor burden that may occur after exposure to a particular treatment. CtDNAs are short DNA fragments (130–145 base pairs) released at low concentration into the circulation, likely by cell apoptosis or necrosis [134]. Due to its recapitulating the genetic mutational spectrum and the epigenetic profile of the originating tumor, ctDNA provides a source of blood-based biomarkers for cancer detection and disease monitoring over time [108]. CtDNA can be isolated from almost any biological fluid [135]. There are currently over 40 commercially available ctDNA extraction methods, including manual and automatic isolation kits. However, these kits use different isolation principles, generally based on the interaction between DNA molecules and magnetic particles, organic compounds, or silica gel membranes, with substantial differences in ctDNA recovery efficiency and size discrimination [136]. These variables may significantly impact the accuracy and reproducibility of a ctDNA analysis experiment, suggesting that the standardization of isolation protocols is imperative to further translate ctDNA into a biomarker in clinical practice. At the time of the conceptualization of this manuscript, there were no studies available regarding the utility of ctDNA in the clinical management of NMSC. There are still only several experiments focused on its applicability in CM.

Currently, serum lactate dehydrogenase (LDH) is the only generally accepted biomarker in the American Joint Committee on Cancer (AJCC) melanoma staging system that can be used to predict CM evolution and indicate therapy efficiency [137]. Elevated LDH is an independent predictor of poor outcomes in patients treated with BRAFi and MEKi [138]; hence, a significant reduction of LDH activity positively correlates with response to immune checkpoint inhibition in metastatic CM patients [139]. Several other circulating proteins, such as S100 and melanoma-inhibiting activity (MIA) protein, have also proved their effectiveness in predicting therapeutic responses in CM; however, they lack sensitivity and specificity and are restricted for use in the clinical setting [19,140]. Nonetheless, a burgeoning body of research suggests that ctDNA may be more consistent and informative for tracking disease status than the traditional serum biomarkers [141,142,143,144,145]. For instance, several ddPCR studies have shown that pre-operative and post-operative ctDNA detection in stage II/III melanoma patients undergoing surgical resection may be associated with increased risk of relapse, potentially informing adjuvant therapy decisions in the affected CM patients [146,147]. Moreover, Shinozaki et al. have shown that the presence of *BRAF* mutations (mt) in ctDNA in the serum of biochemotherapy-treated CM metastatic patients may have clinical utility in predicting tumor response and disease outcome [142]. Briefly, BRAFmts were detected in the ctDNA of 70% of patients in the non-responder group and only 10% of patients within the responder group, being associated with a poor prognosis [142]. In line with this observation, subsequent studies reported that CM patients negative for BRAFmts in ctDNA had longer progression-free survival (PFS) and OS than patients with detectable cfDNA BRAFmt. Syeda et al. reported that CM patients with poor clinical outcomes tended to have increased levels of *BRAF* V600-mt ctDNA before and during treatment (week 4) with dabrafenib or dabrafenib plus trametinib. A ctDNA cutoff point of ≥64 copies/mL, determined via ddPCR, was used to identify patients at high risk for shortened survival PFS (HR = 1.74, *p* < 0.0001) and OS (HR = 2.23, *p* < 0.0001) and ctDNA analysis showed itself to be more informative for disease progression than serum LDH levels [143]. Hence, Sanmamed et al. noted that plasma concentrations of *BRAF*-V600E copies lower than 216 copies/mL were significantly associated with better outcomes as compared with higher concentrations (OS = 27.7 months versus 8.6 months; PFS = 9 months versus 3 months) in CM patients treated with BRAFi [144]. Although it is generally accepted that CM patients negative for BRAFmts in ctDNA have a favorable outcome, differences between threshold values adopted across ddPCR studies may lead to conflicting findings. Therefore, the standardization of the experimental protocols, data analysis, and reporting, is critical and urgently needed, to enable the use of ctDNA in the clinical management of skin cancer patients. For the majority of ddPCR-based ctDNA studies in CM, the concordance between *BRAF* V600 mt in ctDNA and tissue samples was 70–84%, and the sensitivity was 38–79%, suggesting that despite the required analysis optimization, ctDNA may be a promising and robust biomarker for the monitoring of CM patients [141,144]. Finally, there are also several reports highlighting that ddPCR can also be harnessed for the early detection of acquired resistance to targeted therapy in CM. When screening for *BRAF* and *NRAS* variants in a clinical cohort of 48 metastatic CM patients, Gray et al. found circulating *NRAS* mutations in 3 of 7 patients progressing on kinase inhibitor therapy [148]. Reactivation of the MAPK pathway by secondary mutations in *NRAS*, mainly at codon 16 (*p.Q61K/R*), is frequently reported in acquired resistance to dabrafenib/trametinib combination therapy in *BRAF*-mutated CM patients [149]. Notably, NRAS mutations were detected in the ctDNA before the radiological detection of progressive disease, highlighting that the ddPCR-assisted ctDNA mutation analysis may be used to monitor disease evolution and detect the early occurrence of resistance in skin cancer patients [148].

Using ddPCR analysis of ctDNA might also monitor CM patients under immunotherapy. Several research groups pinpointed that the assessment of ctDNA at baseline and during therapy might predict tumor response and clinical outcome in metastatic melanoma patients receiving anti-PD1 antibody therapy. In a study conducted on 76 metastatic CM patients receiving anti-PD1 antibodies, Lee et al. observed that subjects with persistently elevated ctDNA, either at baseline or during therapy, carry a poor prognosis and decreased survival rates [150]. Notably, ctDNA showed increased accuracy over traditional baseline clinical parameters for CM response and prognosis, such as LDH and disease burden [150]. Hence, by employing a ddPCR assay, Seremet et al. confirmed that patients with undetectable ctDNA at baseline have better PFS (HR = 0.47, median 26 weeks versus 9 weeks, *p* = 0.01) and OS (HR = 0.37, median not reached versus 21.3 weeks, *p* = 0.005) rates than patients with detectable ctDNA [151]. In addition, they used a ctDNA cut-off point of 500 copies/mL at baseline and during treatment to stratify the patients as high risk or low risk for disease relapse; thus, ctDNA may be a valuable biomarker for the early identification of tumors refractory to anti-PD1 therapy [151]. Nonetheless, other research groups used ddPCR approaches to distinguish between tumor growth and pseudo-progression, a true challenge in cancer immunotherapy. Pseudo-progression, which consists of an initial increase in the size of tumor lesions, followed by a delayed therapeutic response, occurs in about 10% of immunotherapy treatment cases and is due to the recruitment of various immune cells (e.g., T and B lymphocytes) in the tumor and not due to tumor cell proliferation [152]. This atypical therapeutic response may often be interpreted as a recurrence of the disease, leading, in fact, to the premature discontinuation of an effective treatment [152]. Recently, ddPCR emerged as an accurate tool to differentiate between pseudo-progression and true progression in melanocytic tumors. Lee et al. designed a ddPCR approach to assess the ctDNA levels and mutational status at baseline, and during the first 12 weeks of immunotherapy treatment, in 125 metastatic CM patients [153]. They reported active disease progression in 29 patients (23.2%), but 9 of these 29 patients were soon confirmed with pseudo-progression via imaging assessment. Interestingly, all the nine individuals with pseudo-progression had a significant decrease in, or undetectable, ctDNA levels upon treatment initiation, whereas 18 out of the 20 patients with progressive disease showed no change or a slight increase in their ctDNA levels [153]. All this information suggests that ddPCR-based ctDNA approaches are of high promise in the clinical setting for personalizing the care of immunotherapy-treated skin cancer patients.

LB-based ddPCR approaches also seem clinically feasible to dissect the molecular landscape of brain metastases emanating from cutaneous tumors and to monitor the affected patients. By employing a ddPCR approach, Lee et al. investigated the potential of ctDNA for surveillance and outcome prediction in 72 patients with metastatic melanoma with active brain metastasis under immunotherapy [154]. Thirteen subjects presented with intracranial metastases, whereas the other 59 patients had concurrent intracranial and extracranial metastases and ctDNA detectability at baseline was 0% and 64%, respectively. Detectability was associated with extracranial disease tumor burden [154]. Hence, undetectable ctDNA in therapy correlated with extracranial response but not intracranial response. Patients with undetectable ctDNA at baseline and on-treatment had superior OS than subjects with detectable ctDNA, which suggested that ctDNA might be a robust prognostic biomarker in patients with CM with extracranial metastases. However, ctDNA was inappropriate for studying intracranial disease activity, calling for more refined approaches for monitoring patients with intracranial disease in [154]. Recently, Parietti et al. showed that ddPCR-assisted-detection of a *BRAF* mt in the CSF ctDNA might help assist the early diagnosis of leptomeningeal metastasis as the unique site of CM dissemination, being more effective than both brain magnetic resonance imaging (MRI) and CSF cytology together [43]. Taken together, all this information suggests that CSF ctDNAmt analysis via ddPCR might be an accurate methodology for detecting and monitoring highly aggressive skin melanomas, facilitating therapeutic interventions before identifying the progressive disease by imaging [43].

Another prospective application of ddPCR consists in the analysis of methylated ctDNA. DNA methylation patterns are constantly changing during tumor progression so that DNA methylation analysis can provide relevant clues about the course of the disease and the therapeutic responses [155]. However, it should be noted that ctDNA methylation analysis is highly laborious and challenging, due to the fragmented nature of ctDNA [156]. For this reason, most methylation analysis workflows include an additional step of bisulfite conversion to preserve methylated cytosines available for detection in subsequent molecular analyses [157]. Although several methylation-based ctDNA assays are now commercially available for certain cancers, for CM and NMSC, a methylation-specific ctDNA panel that includes markers of disease progression or drug resistance has not yet been developed [156]. Particularly for CM, efforts are now oriented towards selecting the most appropriate DNA methylation biomarkers among a plethora of hypermethylated and hypomethylated genes constantly identified in melanoma tissues. The primary steps in this direction were taken by Mori et al., who first reported that circulating methylated *RASSF1A* may be a valuable indicator of poor survival and disease refractoriness to bio-chemotherapy in metastatic CM patients [158]. Soon after that, it was found that circulating hypermethylated tumor suppressor genes *PTEN*, *CDKN2A*, and *MGMT* may have diagnostic applications in CM [159]. In parallel, it was reported that methylation levels of transposable element LINE-1 might account for the worse prognosis of stage III CM patients [160]. Although all these biomarkers are speculative and not yet validated for use in the clinical setting, it would be interesting to test them through ddPCR for their clinical relevance in the prognosis and monitoring of CM patients.

Despite all these promising results, several limitations need to be overcome to allow the use of ctDNA in the clinical management of skin cancer. These limitations may include, but are not limited to, the following: variations in specificity and sensitivity among different detection approaches, lack of harmonization and standardization of experimental protocols that introduce biases and prevent obtaining robust data, as well as high economic costs [161]. The probability of getting false-negative results may be another limitation when interrogating ctDNA intended as clinical information in cancer patients. It should be noted that although ctDNA detection may be a valuable indicator of tumor burden throughout disease progression, there may also be certain metastatic patients developing tumors that do not shed ctDNA into the circulation, so, in these cases, the ctDNA mutations/fragments could go undetectable regardless of the ddPCR’s sensitivity [162,163]. However, using mitochondrial tumor-derived DNA as an alternative source of ctDNA might help overcome these limitations, given that it has been reported that there are thousands of copies of mitochondrial DNA per cell [163,164]. The volume of plasma yielded from a typical blood sample of 10 mL may also impact the accuracy of ctDNA analysis, as a low volume of plasma can limit the number of available genome copies to be analyzed and, subsequently, the accurate detection of variants at low allele frequency [165]. Moreover, metastatic tumors confined to specific secondary sites, such as the central nervous system (CNS), may also release lower amounts of ctDNA into the bloodstream, providing conflicting information on disease status [166]. In the scenario of cerebral metastasis, the molecular profiling of ctDNA seems challenging since the blood-brain barrier prevents it from entering the circulation. Consequently, alternative non-blood sources of ctDNA, such as CSF, urine, sputum, or stools, may be considered when monitoring affected patients for disease recurrence and therapy response [163].

At the moment, more than 20 clinical trials are underway to study either the prognostic value of BRAF- or NRAS-mutated ctDNAs, their dynamics during treatments, and the standardization of experimental methods for ctDNA quantification and mutation detection in various types of tumors [135]. Therefore, although there is still a long way to go, ctDNA is likely to someday find its place in the clinic, where it can provide valuable information on tumor load, disease progression, and survival outcomes irrespective of the tumor genotype in skin cancer patients. Finally, whereas the significance of ctDNA is more prominent in the clinical management of advanced skin cancer patients, its clinical utility in patients with no evidence of disease, or with early-stage disease, remains largely unknown [135].

### 5.2. Circulating miRNAs Analysis

DdPCR may also be a promising approach for analyzing the circulatory miRNAs in patients’ body fluids. As previously mentioned, miRNAs are critical regulators of gene expression in both health and disease. MiRNAs have been proven to be dysregulated in cancerous samples, and many data are currently available regarding their potential applications as prognostic and predictive biomarkers [167,168,169]. As in any other cancers, in skin cancers circulating miRNAs can be found freely circulating free, complexed with proteins or encapsulated in vesicles, such as exosomes [170]. Hence, they present increased stability even outside the cell, are tissue-specific, and vary with disease states, being informative for disease progression or therapeutic responses. All these characteristics make them promising blood-based biomarkers [115]. MiRNAs can be isolated from biological fluids via simple phenol-chloroform-based extraction methods or phenol-chloroform-based phase separation coupled with column-based clean-up techniques. At the moment, at least 15 types of circulating miRNAs isolation kits are commercially available, but they show significant differences in terms of performance and assay reproducibility [171]. Hence, normalization remains one of the most challenging aspects of digital and qPCR data analysis, since no generally accepted normalization strategies and reference genes exist for circulating miRNA quantification [172]. Nonetheless, although further research is needed around this topic, it is expected that the conceptualization and dissemination of specifical guidelines on standardized pre-analytical and analytical procedures, and on specific normalization methods, would increase the reproducibility of research results so that miRNA expression levels could be compared between studies and successfully used for clinical purposes [172].

The majority of publications regarding circulating miRNA applications focus on CM, probably due to its highly metastatic potential, increased resistance to standard therapies, and elevated genomic heterogeneity [170]. Circulating miRNA signatures have proven their clinical utility in differentiating CM from other skin cancers and for diagnosis, prognosis, and predictive applications. For instance, specific molecules identified through conventional methods (qPCR) in the sera of CM patients, such as miR-150-5p, miR-149-3p, miR-193a-3p, miR-15b-5p, and miR-524-5p, have been shown to play essential roles in CM prognosis and diagnosis [173,174]. Furthermore, miR-214 and miR-579-3p levels emerged as robust biomarkers of response to targeted therapy resistance in CM patients [175,176]. Yet, a recent study has shown a panel of circulatory miRNAs (e.g., miR-146a, miR-155, miR-125b, miR-100, let-7e, miR-125a, miR-146b, and miR-99b) that might be related with myeloid-derived suppressor cells (MDSCs) and ICIs resistance in CM patients [177]. Briefly, these molecules were found to regulate the transcriptional reprogramming of the myeloid cells towards an MDSC phenotype, which exerts immunosuppressive functions in tumors. Therefore, in addition to their roles as predictive biomarkers, miRNAs might also serve as valuable therapeutic targets for pharmacological interventions in CM [177]. Currently, reports regarding the assessment of circulating levels of miRNAs in skin cancer patients via ddPCR are scarce; there is just a single report on melanoma cell lines [178]. However, the results achieved with ddPCR assays for circulating miRNAs in other tumor types are encouraging, culminating in identifying biomarkers relevant for oral cancer, ovarian cancer and esophagogastric cancer clinical management [32,179,180]. Consequently, ddPCR methods are expected to become part of biomedical and translational skin cancer research, holding promise for more accurate and personalized approaches for these patients.

### 5.3. CTCs Analysis

Besides ctDNA and miRNAs, circulating tumor cell (CTC) characterization via ddPCR may offer novel insights into tumor invasion and therapeutic responses in skin cancer patients. Dissociated from the primary tumors or metastases, CTCs are rare cells present in the blood and lymphatic circulation that allow for both mRNA and DNA genomic profiling [181]. The identification of a high number of CTCs in patients with solid tumors, including CM and SCC, predicts poor survival, most likely due to CTCs’ ability to act as “seeds” of clinical metastases [182,183].

There are many CTC isolation techniques; however, they may be broadly divided into biochemical and biophysical approaches, depending on their principle [184]. The former method relies on identifying specific molecular biomarkers of CTCs, and one such example is the immunomagnetic-based assay targeting EpCAM proteins. At the same time, the latter differentiates CTCs from other blood cells, based on their physical properties [184]. Nonetheless, the development of CTC isolation techniques incorporating microfluidic systems has demonstrated better performances in CTC isolation compared with conventional CTC isolation assays. The magnetic CellSearch^®^ Circulating Melanoma Cell Kit, the dual-step dielectrophoretic cell separation technology DEPArray, and the new in vivo photoacoustic flow cytometry platform “Cytophone” are the most important microfluidic chips and biosensors that have revolutionized CTC capture in skin cancer [135]. However, until recently, the molecular characterization of CTCs seemed very challenging, due to the need for more sensitive methods to detect and quantify the subtle omic alterations present at this level. Nonetheless, the development of ddPCR technology has largely solved this issue, providing reliable translational information on tumor progression, therapeutic responses, and survival outcomes for skin cancer patients and others [181].

Various ddPCR methodologies have been proposed and developed for the molecular profiling of CTCs. For instance, several research groups used ddPCR to detect specific mutations in the CTCs of cancer patients. One such study, conducted by Reid et al., showed that ddPCR might be successfully employed to screen for *BRAF*-V600E or V600K mt in CTC-based LBs harvested from metastatic melanoma patients [185]. *BRAF*-V600E and V600K mts were detected in 77% and 44%, respectively, of enriched CTC fractions of metastatic CM patients with recorded mutated tumor tissues. Moreover, the authors reported that ddPCR was 200 times more sensitive than competitive allele-specific PCR (castPCR), allowing the detection of *BRAF* V600E/K mt down to frequencies of 0.0005% [185]. Therefore, the identification of actionable mutations in CTCs via ddPCR may be a promising strategy for monitoring disease progression and predicting failure before clinical relapse in CM patients. In parallel, Denis et al. employed ddPCR to screen for KRASmt in the CTCs from CRC patients [186]. KRASmt, which are negative predictors of *EGFR*-targeted antibody treatment efficiency in CRC, were identified in 30/35 samples. The researchers found a correspondence rate of 77% of *KRAS* genotyping between CTCs and the corresponding tumor tissues, and a sensitivity of 83% [186]. Therefore, the minimally invasive nature of LBs coupled with the sensitivity of ddPCR might soon provide outstanding opportunities related to diagnosis and monitoring for the clinical management of cancer patients, including those diagnosed with skin malignancies.

Apart from its prospective roles in quantifying hotspot mutations in CTCs, the ddPCR may also enable the quantification of CTC-derived transcripts, which are of great interest in assessing minimal residual disease (MRD) and therapeutic responses in cancer patients. One of the most relevant transcripts to be evaluated are those corresponding to tumor-associated antigens (TAAs) (e.g., MAGE-A3, PAX3, and MART-1), which are reliable indicators of immunotherapeutic responses in CM patients [187]. Nonetheless, ddPCR assays for TAA evaluation are currently employed in hematological diseases to address tumor heterogeneity and predict patient outcomes [188]. In addition to TAAs, other CTCs-associated transcripts may also be of clinical relevance in cancers. For instance, Hong et al. showed that digital RNA-based quantitation of 19 melanoma CTC-derived transcripts enabled the non-invasive monitoring of CM patients on immunotherapy [189]. Patients that experienced a decline in the CTC score at seven weeks had better OS; in contrast, a rise in CTC score was associated with therapeutic failure and poor survival in 53% of the affected patients. Therefore, in the absence of other blood-based biomarkers, the assessment of CTC-transcriptomic neural crest signature in CM may be a promising approach to distinguish patients at high risk of disease relapse following ICIs, with increased accuracy and minimal invasiveness [189]. Finally, there are several reports that present the development and analytical validation of ddPCR assays that allow for PD-L1 quantification in CTCs [190,191]. In many cancers, including advanced melanoma, PD-L1 expression on CTCs may be predictive of response to ICIs and longer PFS [192]. Accordingly, ddPCR-assisted PD-L1 quantitation in CTCs may be an effective strategy to identify skin cancer patients who are most likely to benefit from pembrolizumab at an early stage of treatment, as well as to obtain mechanistic insight with respect to immunotherapy resistance mechanisms.

Still, some limitations need to be addressed before translating CTCs into the clinical management of skin cancer patients. For instance, the great variety of molecular markers used for CTCs enrichment, usually with low sensitivity, and the diversity of technical procedures used for CTC isolation may affect the clinical relevance of the obtained results [135]. Furthermore, considering the rare presence of CTCs within the blood circulation and the remarkable heterogeneity of CTC phenotypes and functional states, relative to the tissue of origin, the robustness of scientific findings on CTCs remains questionable [193]. However, recent progress in CTC enrichment and isolation technologies [135] and specific clinical trials have the potential to help solve these issues [194,195,196].

### 5.4. EVs Analysis

DdPCR may also enable the identification of cancer-associated mutations in extracellular vesicles (EVs), which may be a promising strategy for validating novel biomarkers in cancers, including skin cancer. Although initially identified by Pan and Johnstone in reticulocytes and considered a waste disposal mechanism [197], EVs have emerged as key mediators of intercellular communication within the TME, mainly through their ability to transfer their biological content between cells [198]. There are at least four types of EVs: microvesicles (MVs), exosomes, oncosomes, and apoptotic bodies, which differ in terms of their biogenesis, dimensions, and associated biomarkers [199]. Oncosomes and apoptotic bodies are among the larger EVs, with a diameter ranging from 1 to 10 μM [200,201]; in contrast, MVs and exosomes are smaller EVs with diameters ranging from 100 to 1000 nm and 30 to 100 nm, respectively [200]. Except for the apoptotic bodies, all EVs are released into circulation by living cells, either by direct budding of the plasma membrane (oncosomes and microvesicles) or via the endosomal pathway (exosomes) [200,202]. EVs can be found in many biological fluids, including blood [203], amniotic fluid [204], urine [205], CSF [206], breast milk [207], saliva [208], ascites [209], tears [210], semen [211], bronchoalveolar lavage fluid [212], and in the conditioned media of cultured cells [213].

Nonetheless, EVs are cup-shaped nanovesicles encapsulating a plethora of molecular constituents, such as cytoskeletal, transmembrane, and thermal shock proteins, lipids, enzymes, DNA, and heterogeneous species of RNA, which may be relevant for the molecular profiling of human cancers [214]. According to the most current versions of the Vesiclepedia database, which centralizes information from 1254 studies, at least 349,000 proteins, over 27,000 mRNAs, and over 10,000 miRNAs have been identified into the composition of EVs [215]. The EV cargo depends on the cell of origin and can be trafficked between cancer cells or cancer cells and components of TME, modulating their intracellular signaling pathways, gene expression, and phenotypes. EVs are stable under various storage conditions, revealing bright perspectives toward more personalized diagnosis, monitoring, and therapeutic approaches in cancers [135]. There are many procedures for EV isolation, such as differential ultracentrifugation (UC), precipitation, immunoaffinity capture, ultrafiltration, size exclusion chromatography, and microfluidic platforms [216]. Hence, EVs’ stability may be considerably impacted by multiple pre-analytic factors, such as sample storage time, temperature, anticoagulants, and centrifugation parameters [216]. Aiming to overcome the challenges associated with the variability in the methodology for sample collection, storage, and analytical methods across multiple studies, the International Society for Extracellular Vesicles (ISEV) has initiated several workshops to achieve a consensus on best practices and acknowledge the potential obstacles in translating EVs in clinical applications. They highlighted the need for uniformized technologies for EV isolation and characterization and generally accepted reference materials [217]. Therefore, as concerted efforts are oriented in this direction, EVs might soon be translated into routine clinical practice, paving the way for personalized approaches in oncological diseases.

Finally, several other research studies highlighted that ddPCR might be successfully employed to detect cancer-specific biomarkers on EVs from the plasma of skin cancer patients, with unprecedented resolution, compared to traditional approaches (Figure 3). For instance, Zocco et al. developed a ddPCR protocol to assess the benefit of detecting *BRAF* V600mt in EV-DNA in addition to ctDNA in metastatic melanoma patients at the beginning and during BRAFi therapy [218]. It is also worth mentioning that they employed an ultracentrifuge-free assay to isolate plasma EV-DNA and ctDNA. Similar to other reports, the authors found that *BRAF* V600E copy levels above 50 copies/mL of plasma in ctDNA and EV-DNA correlated to poorer prognosis and OS rates. Hence, they reported that the dynamics of *BRAF* V600E copy numbers might be relevant for monitoring the response to BRAFi in CM patients. *BRAF* V600E copy levels were almost undetectable after exposure to BRAFi, but further increased when the tumors acquired drug resistant-phenotypes [218]. The screening of cancer-derived EVs via ddPCR also proved useful in deciphering the molecular mechanisms associated with drug resistance in skin cancers. The expression of aberrantly spliced *BRAF* V600E isoforms is one of the most frequently reported mechanisms of resistance in melanoma patients progressing on BRAFi [219]. Four *BRAF* splicing variants have been described, based on their predicted molecular weight, namely p61, p55, p48, and p41 [219]. Interestingly, Clark et al. developed a custom ddPCR assay for the presence of *BRAF* splicing variants in plasma cell-free RNA (cfRNA) from CM patients [220]. Notably, 24 of 38 patients who experienced disease progression following *BRAF*/MEK inhibition showed an increase in ctDNA levels at the time of relapse. Hence, circulating *BRAF* splicing variants were detected in cfRNA from 3 of these 38 patients; two presented with the *BRAF* p61 variant and one with the p55 variant. Isolation and analysis of RNA from EVs from resistant cell lines and patient plasma showed that *BRAF* splicing variants are associated with EVs. These findings indicate that in addition to plasma ctDNA, RNA incorporated in EVs can provide specific information on tumor progression in real-time [220]. In parallel, Yap et al. reported that several patients might present mutations in EVs that are undetectable in tissue, suggestive of emerging resistance to targeted therapy prior to radiological evidence of tumor progression [221].

## 6. Discussion

Skin cancer is the most common neoplasm in Caucasians [1]. Skin cancer dramatically affects quality of life, as it can be disfiguring or even deadly. Cutaneous tumors can evolve from keratinocytes cellular components and, hence, are non-melanoma skin cancer (NMSCs) that sub-divide in basal cell carcinoma (BCC) and squamous cell carcinoma (SCC), or they can evolve from melanocytes and lead to cutaneous melanoma (CM) and, last, but not least, they can still have an indistinct cellular origin and lead to Merkel cell carcinoma (MCC) [6]. According to the most recent GLOBOCAN estimates, there were more than 320,000 new cases of CM worldwide in 2020, which resulted in 57,000 deaths and about 1.2 million new cases of NMSC [5]. In the last two years, the COVID-19 pandemic, which has become the epicenter of daily clinical practice, restricted access to healthcare facilities and delayed the diagnosis of patients with CM and other skin cancers, resulting in increased rates of morbidity, mortality, and, consequently, a greater financial burden on the health system [12]. CM is fatal if diagnosed at advanced stages, while the keratinocyte–derived cancers, SCC and BCC, are generally curable but debilitating and disfiguring [4]. Given the poor prognosis of advanced-stage skin cancers, there is an urge to find more reliable biomarkers for early diagnosis, prognosis, and treatment response in these patients.

DdPCR has emerged as one of the most accurate and sensitive tools for examining omics alterations in various tumors, including skin cancers. The ddPCR method can be successfully applied for absolute allele quantification, rare mutation detection, CNV analysis, DNA methylation, and transcriptomic evaluation in various types of biological samples [133]. This methodology has proved beneficial for FFPE tumor tissue analysis, where limited sample availability and inferior DNA quality are challenging for the majority of molecular assays. However, within the last years, most applications of ddPCR in cancers have focused on LBs, including ctDNA and ctRNA, EVs, and CTCs [77]. In contrast to highly invasive and spatially limited tissue biopsies, LBs allow for repeat sampling and longitudinal monitoring of disease progression over time, providing outstanding opportunities for the detection of MRD or therapy resistance, as well as recurrence or disease progression [222]. Nonetheless, research presented at the American Society of Clinical Oncology (ASCO) meetings highlighted how advances in dermato-oncology assisted by ddPCR and LBs may inform future research and clinical decisions in cutaneous tumors [223,224,225]. Among all skin cancers, the most abundant studies regarding ddPCR assays have been developed in CM, probably due to its highly metastatic potential, increased recurrence, tolerance to systemic therapies, and high genomic heterogeneity.

Several studies demonstrated that ddPCR assays might play a valuable role in the accurate diagnosis and prognosis of skin cancers. Reinders et al. showed that the sensitivity of ddPCR may be harnessed to detect low-grade post-zygotic mosaicism of *PTCH1* gene mutations in patients suspected of BCNS, even if they do not present with specific clinical manifestations [87]. Since in the case of post-zygotic mosaicisms, the clinical manifestations may be less or more visible, depending on the tissues involved and the mutational load, identifying patients at risk for BCC, as well as screening them as carriers via ddPCR, seems to be a promising strategy for improved cancer prevention and genetic counselling in the affected patients [87]. In parallel, other studies highlighted that ddPCR should be the primary method for detecting and monitoring *BRAF* V600E-mutant melanomas. DdPCR showed enhanced sensitivity in detecting the oncogenic *BRAF* V600E mt compared with conventional methods such as qPCR, Sanger sequencing, AS-PCR, or pyrosequencing, being able to identify it in several archival tissues when the others could not detect it [68,91]. Notably, more than half of the patients who tested *BRAF* V600E positive only through ddPCR presented later with sentinel lymph node metastases, suggesting that ddPCR is the most suitable methodology for detecting the low-frequency *BRAF* V600E-positive melanoma clones in patients’ tissues [91]. However, the identification of *BRAF* mutations alone in LBs is not yet validated as a CM early detection strategy since these mutations have also been reported in normal and pre-neoplastic skin clones [226].

Although tissue biopsy represents the gold standard for diagnosis and treatment choice in cancer, LB is currently regarded as a promising non-invasive method that may complement tissue biopsy in clinical practice. However, considering the data available at the moment, the use of ddPCR-based LB analysis for diagnostic purposes is not a priority in the field of dermato-oncology, probably due to the identification of certain low DNA-shedding tumors, which may lead to ambiguous findings in the affected patients [227], and increased accuracy of reflectance confocal microscopy (RCM) and optical coherence tomography (OCT) that are currently employed in the clinical setting for non-invasive skin cancer diagnosis [228]. Hence, LB analysis seems to be more appropriate for monitoring high-risk skin cancer patients with advanced-stage disease, particularly to assess their disease progression and response to therapy, enabling early adaptive changes to a patient’s treatment if necessary [135].

A robust body of evidence suggests that ddPCR is an ideal methodology for analyzing ctDNA in skin cancer patients. Several studies demonstrated the prognostic value of ctDNA assessed via ddPCR, as high levels of *BRAF* V600-mutated ctDNA at melanoma diagnosis correlated with shorter PFS and OS rates [229]. In addition, the increase of ctDNA bearing the same mutation might reflect disease progression in melanoma patients under BRAFi ± MEKi, preceding the radiological detection of the tumor [27,143]. Furthermore, ctDNA negativity assessed via ddPCR seemed to be a good predictor of response to CTLA-4 or PD-1 inhibition in patients with advanced-stage disease [150,151]. Hence, assessing the efficacy of immunotherapeutic regimens with anti-CTLA-4 or anti-PD-1 agents is challenging for oncologists due to pseudo-progression. This delayed immune response inaccurately indicates a refractory disease; however, in several studies, ctDNA negativity at progression was attributable to potential pseudo-progression events, highlighting that ddPCR-evaluated ctDNA levels may help predict patients’ clinical outcomes following immunotherapy [153]. It is also worth mentioning that plasma ctDNA is not suitable for monitoring disease progression in CM patients with intracranial metastases, and MRI remains the exclusive surveillance modality for these patients. Nonetheless, recent studies have shown that ctDNA detectable in the CSF of CM patients with brain metastases may be a valuable surrogate biomarker in such situations [43]. Finally, the evaluation of methylation in ctDNA via ddPCR, such as the detection of *RASSF1A* or paraoxonase 3 (PON3), may suggest the state of the disease and survival outcomes in CM patients, even in the absence of tumor mutation data for *BRAF*, *RAS* or *EGFR* genes [159,230]. Despite all these promising results, several limitations need to be overcome to allow the use of ctDNA in the clinical management of skin cancer. These limitations may include, but are not limited to, the following: variations in specificity and sensitivity among different detection approaches, the lack of harmonization and standardization of experimental protocols that introduce biases and prevent the obtaining of robust data, as well as considerable economic costs [161]. The probability of getting a false-negative result may be another limitation when interrogating ctDNA via ddPCR for clinical information in cancer patients. It should be noted that although ctDNA detection may be a valuable indicator of tumor burden throughout disease progression, there may also be certain metastatic patients with tumors that do not shed ctDNA into the circulation, so in these cases, the ctDNA mutations/fragments could go undetected, regardless of ddPCR’s sensitivity [162,163]. Therefore, all these shortcomings suggest that the combination of multiple detection strategies, including tissue biopsies, liquid biopsies, and imaging tests, is most appropriate to provide a comprehensive view of the molecular landscape of a tumor and its relationship to therapeutic responses and disease evolution. Despite their remarkable sensitivity and specificity, ddPCR is a relatively young technology and cannot entirely decipher the omic complexity of heterogeneous tumors, and, consequently, cannot fully replace the traditional approaches [133]. At the moment, more than 20 clinical trials are underway to study the prognostic value of *BRAF*- or *NRAS*-mutated ctDNAs, their dynamics during treatments, and the standardization of experimental methods for ctDNA quantification and mutation detection [135]. Therefore, although there is still a long way to go, ctDNA is likely to someday find its place in clinical management, where it could provide complementary information on tumor load, disease progression, and survival outcomes, irrespective of the tumor genotype in skin cancer patients.

The ddPCR method may also be used to characterize CTCs in skin cancer patients. In particular, for CM, the assessment of a specific CTC-transcriptomic neural crest signature may be a promising approach to distinguish patients at high risk of disease relapse following ICIs, with increased accuracy and minimal invasiveness [189]. There are also several reports presenting how ddPCR assays allow for PD-L1 quantification in CTCs [191]; notably, the expression of PD-L1 on CTCs has been shown to predict the response to pembrolizumab in advanced melanoma patients [231]. Finally, ddPCR may enable the quantification of actionable mutations in CTCs in CM [185]. Still, some limitations need to be addressed before translating CTCs into the clinical management of skin cancer patients. For instance, the great variety of molecular markers used for CTC enrichment, usually with low sensitivity, and the diversity of technical procedures used for CTC isolation may affect the clinical relevance of the obtained results [135]. Furthermore, considering the rare presence of CTCs within the blood circulation and the remarkable heterogeneity of CTC phenotypes and functional states relative to the tissue of origin, the robustness of scientific findings on CTCs remains questionable [193]. However, recent progress in CTC enrichment and isolation technologies [135] and specific clinical trials will soon help solve these issues [194,195,196].

Moreover, ddPCR also proved helpful in analyzing other circulating biomarkers, including circulatory miRNAs and EVs. However, the lack of consensus regarding methodologies used to detect and quantify these circulating biomarkers prevents validating a specific panel for use in clinical practice at the moment [135]. DdPCR has shown increased accuracy in detecting *BRAF* p61 and p55 splicing variants in EV-RNAs, providing vital information on CM resistance to targeted therapy [220]. Hence, other authors have reported that ddPCR analysis of EV-derived RNAs may reveal several mutations undetectable in tumor tissues, highlighting that this strategy might inform clinicians about the occurrence of drug resistance in CM patients before radiological identification of the progressive disease [221].

Therefore, analyzing archival tissues or LBs, ddPCR provides outstanding opportunities for skin cancer screening, prognosis, detection of MRD, monitoring, and treatment selection, serving as a platform for personalized medicine in this heterogeneous disease.

## Figures and Tables

**Figure 1 jpm-12-01136-f001:**
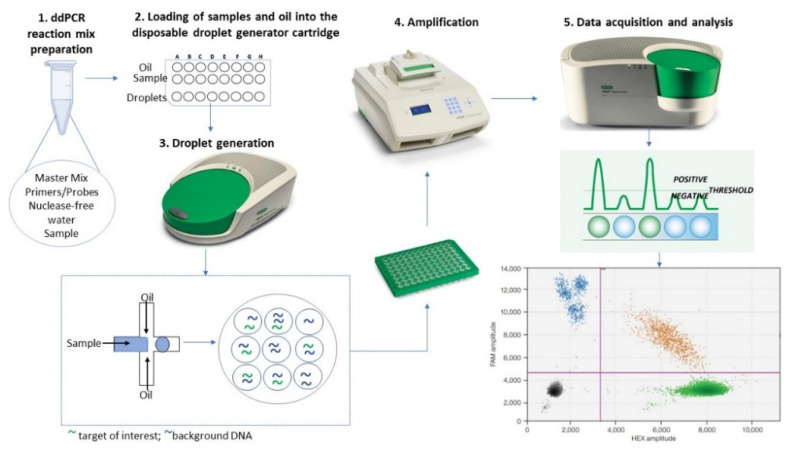
Schematic representation of a ddPCR assay.

**Figure 2 jpm-12-01136-f002:**
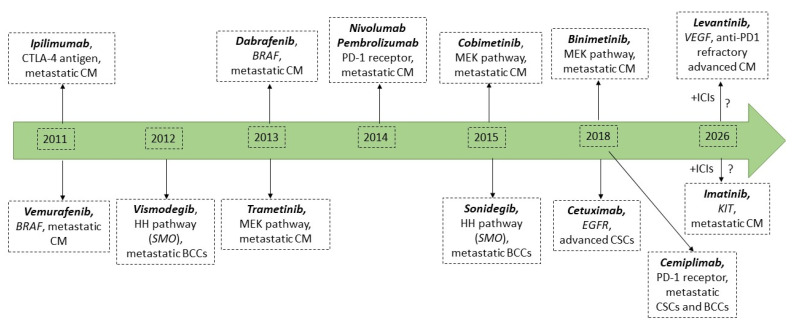
Timeline depicting genomic biomarker-driven drug approvals in skin cancer. Except for Imatinib, a small-molecule inhibitor of the KIT tyrosine kinase, and Levantinib, a multi-kinase inhibitor of the vascular endothelial growth factor (VEGF) receptors, which are currently tested in clinical trials for their efficiency when combined with ICIs, all the other drugs have gained FDA approval for use in the clinical setting in skin cancer patients. CTLA-4- Cytotoxic T lymphocyte antigen 4; ICIs- immune checkpoint inhibitors; HH pathway- Hedgehog signaling pathway; SMO- Smoothened, Frizzled Class Receptor; MEK pathway- Mitogen-activated protein kinase kinase pathway; PD-1 receptor- Programmed cell death protein 1; EGFR- Epidermal growth factor receptor; VEGF- Vascular endothelial growth factor; KIT- KIT Proto-Oncogene, Receptor Tyrosine Kinase.

**Figure 3 jpm-12-01136-f003:**
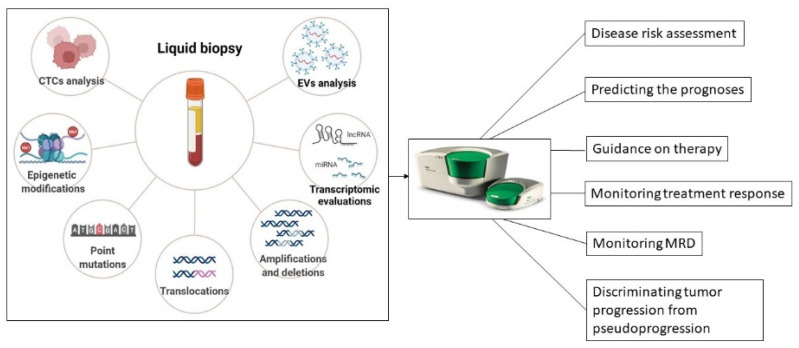
Clinical applications of liquid biopsy in the management of skin cancers.

**Table 1 jpm-12-01136-t001:** Comparison of different omics technologies used for the routine molecular testing of tumors ^1^.

Technology	Assay	Sensitivity	Specificity	LoD	Type of Alterations	Strengths	Limitations	Ref.
Real-time PCR	AS-PCR	1%	98%	0.001%	Know point mutations (SNVs, Fusions, Indels CNVs)	Ease of design and execution; High sensitivity and specificity of detection with fluorescent hydrolysis probes; No need for informatics expert support.	Detects only known genomic variants in limited genomic regions; Reduced multiplexing capability; Quantitation requires standard curve using appropriate positive controls.	[75,76]
	MS-PCR	0.62%	89–100%	0.1%	Known methylation sites	Ease of design and execution; Increased sensitivity when analyzing small quantities of methylated DNA; No need for informatics expert support.	Detects only specific CpG islands.	[76]
ddPCR		0.001–0.1%	100%	0.005%	Know point mutations (SNVs, Fusions, Indels, CNVs)	Absolute quantitation possible because of scanning and Poisson-based counting of droplets; No need for a standard curve for quantitation; Short turnaround time; No need for informatics expert support.	Unsuitable for mutation screening and identification of novel variants; Reduced multiplexing Capability.	[68,75,76]
NGS	WGS	5–10%	80–99.9%	5–10%	Genome-wide CNVs, DNA methylation studies	Prior knowledge of mutations not required; Genome-wide profiling; Identification of specific cancer signatures. Pathogenic gene screening; Detection of CNVs, fusion genes, rearrangements, neoantigens and TMB.	Extensive bioinformatics support; Variable sensitivity and specificity (increase depth leads to higher costs); Long turnaround time; Costly and not appropriate for patient longitudinal monitoring.	[76]
	WES	5%	80–95.6%	5%	Coding regions, gene promoters, intron-exon junctions, non-coding DNA of miRNA genes			
	TargetedNGS gene panels	0.01–0.1%	99.6%	2–5%	Know point mutations	Increased sensitivity and specificity compared to WES/WGS; Produces a smaller and more manageable data set compared to untargeted approaches, making analysis easier.	Less comprehensive than WES/WGS; amplicon methods based on multiplex PCR.	[74]
Sanger sequencing		15–20%	100%	20–25%	Know point mutations	Provides sequence information and determines whether a point mutation or small deletion/duplication is present.	Low sensitivity; Low discovery power; Costly and laborious.	[78]

^1^ LoD-limit of detection; AS-PCR-allele-specific real-time PCR; MS-PCR-Methylation-specific PCR; SNVs-single-nucleotide variants; CNVs-copy number variation; NGS-Next-generation sequencing; WGS-Whole genome sequencing; TMB-Tumor mutational burden.

## Data Availability

Data sharing is not applicable.

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
