# Peer review of "Skin Cancer Research Goes Digital: Looking for Biomarkers within the Droplets"

_jpm, 2022, doi:10.3390/jpm12071136_

Round 1
Reviewer 1 Report
I read with interest this review manuscript regarding digital droplet PCR applications in skin cancer.
The aims were to summarise the available evidence in the applications of ddPCR for skin cancers. However, in an attempt to summarise ddPCR evidence, it has ignored the evidence more widely for liquid biopsies using other technologies. While ddPCR remains a tumour agnostic, cheap and has a quick turnaround, there are significant drawbacks which are downplayed in the discussion. For example, there is no mention of PCR associated sequencing errors which can be a major problem. The discussion presented is almost one-sided to ddPCR.
Most large scale early detection strategies (especially when pan-cancer) are employing epigenetic (methylation, fragmentomics etc) and mutation-based combination strategies to get to the low variant allele fractions required to be useful. The false positive rates here are really important to consider in terms of wider usage.
Minor criticisms
1. Page 2 Line 68 : NRAS not a directly actionable mutation
2. Page 8 Line 334- : I am surprised at the downplaying of the role of clinicopathological factors - this is extremely surprising as they are currently the only validated prognostic marker of long term clinical behaviour.
3. Discussions around BRAF and BRAF variants (such as V600K) - this ignores the fact that BRAF mutations are common in normal and pre-neoplastic skin clones, and therefore identification of BRAF mutations alone in liquid biopsies is not yet validated as an early detection strategy. The data on BRAF variant outcomes are based on small numbers in subselected populations, and the treatment outcome data again is based on post hoc analysis. I would be wary of overemphasising this.
4. Similarly, KRAS-associated melanoma outcomes are not widely validated to the extent of making long term prognostic claims.
5. I believe the reference list to be too long (250 seems excessive, especially as many references do not add to the main arguments made)
6. There are several grammatical errors which I have not highlighted in detail - please review
Author Response
Dear Reviewer,
We want to thank you for the thorough reading of this manuscript and your thoughtful recommendations, which greatly help revise the manuscript.
We agree with your comments and have revised our paper accordingly.
Q1: However, in an attempt to summarise ddPCR evidence, it has ignored the evidence more widely for liquid biopsies using other technologies. While ddPCR remains a tumour agnostic, cheap and has a quick turnaround, there are significant drawbacks which are downplayed in the discussion. Most large scale early detection strategies (especially when pan-cancer) are employing epigenetic (methylation, fragmentomics etc) and mutation-based combination strategies to get to the low variant allele fractions required to be useful. The false positive rates here are really important to consider in terms of wider usage.
A1: We sincerely appreciate the reviewer’s comments. As suggested, we have mentioned the currently available technologies for ctDNA analysis besides PCR. The discussions concerning NGS approaches for LBs analysis can be found between lines 263-269 and lines 297-318, respectively. Consequently, we have done several minor modifications to table 1 to include more details on the advantages and limitations of targeted and untargeted NGS assays in cancer research.
We also highlighted the analytical limitations of ddPCR technology, especially when used as an early detection strategy in oncology. The PCR-associated sequencing errors are discussed as well. These discussions can be found within the revised manuscript, lines 263-296.
Q2: Page 2 Line 68 : NRAS not a directly actionable mutation
A2: Thank you so much for your comment. BRAF V600E/K are the sole mutations amenable to BRAFi and MEKi in CM. I have corrected it in the text (line 68).
Q3: I am surprised at the downplaying of the role of clinicopathological factors - this is extremely surprising as they are currently the only validated prognostic marker of long term clinical behaviour
A3: We have added several considerations regarding the role of clinicopathological factors as prognostic biomarkers in skin cancer. They start at line 460.
Q4: Discussions around BRAF and BRAF variants (such as V600K) - this ignores the fact that BRAF mutations are common in normal and pre-neoplastic skin clones, and therefore identification of BRAF mutations alone in liquid biopsies is not yet validated as an early detection strategy. The data on BRAF variant outcomes are based on small numbers in subselected populations, and the treatment outcome data again is based on post hoc analysis. I would be wary of overemphasising this.
A4: The authors thank you for the suggestion. We have added several explanations into the manuscript as follows:
-lines 444-456; lines 1092-1094: added information on BRAF mutations occurrence in common in normal and pre-neoplastic skin clones
-lines 1092-1094: emphasized that identification of BRAF mutations alone in liquid biopsies is not yet validated as an early detection strategy in skin cancer
- lines 522-526: highlighted that the data on BRAF variant outcomes are based on small numbers in subselected populations, and the treatment outcome data again is based on post hoc analysis
Q5: Similarly, KRAS-associated melanoma outcomes are not widely validated to the extent of making long-term prognostic claims.
A5: Yes, we have addressed it in the manuscript, lines 531-536.
Q6: I believe the reference list to be too long (250 seems excessive, especially as many references do not add to the main arguments made)
A6: Thank you so much for your valuable suggestion. As suggested, we eliminated the unnecessary references, and now the total number is 231.
Q7: There are several grammatical errors which I have not highlighted in detail - please review
(x) Moderate English changes required
A7: We regret there were problems with the English. A native English speaker has now carefully revised the paper to improve the grammar and readability.
We have also corrected several typo errors such as NRAF- right: NRAS (line 832), and counselling- right: counseling (line 414).
We hope that you find our responses satisfactory and that the manuscript is now acceptable for publication.
Kind regards,
Authors of jpm-1778449
Reviewer 2 Report
In this manuscript, the paragraphs are well written. The topic is good and the article could be improved by these minor editing;
1. Explain the difference between ddPCR and real-time PCR for detecting cancer biomarkers. More explanation (in section 2 or 3) is needed about the superiority of ddPCR over real-time PCR for cancer diagnosis.
2. In the section 5 of the manuscript, wherever words such as. ctDNA, CTC, EV,..are written for the first time, they must be spelled out in full term.
Author Response
Dear Reviewer,
We want to thank you for the thorough reading of this manuscript and your thoughtful recommendations, which greatly help revise the manuscript. We are also grateful for the positive feedback that we have received.
We agree with your comments and have revised our paper accordingly.
Q1: Explain the difference between ddPCR and real-time PCR for detecting cancer biomarkers. More explanation (in section 2 or 3) is needed about the superiority of ddPCR over real-time PCR for cancer diagnosis.
A1: We sincerely appreciate the reviewer’s comments. As suggested, we have included several explanations on the superiority of ddPCR over real-time PCR for cancer diagnosis in section 2 of our manuscript, lines 194-226.
Q2: In the section 5 of the manuscript, wherever words such as. ctDNA, CTC, EV,..are written for the first time, they must be spelled out in full term.
A2: Thank you so much for your comment. We have added to the text the explanations of the following abbreviations: ctDNA (line 661), CTCs (line 889), and EVs ( line 967).
We hope that you find our responses satisfactory and that the manuscript is now acceptable for publication.
Kind regards,
Authors of jpm-1778449